# Quantitation of Flavor Compounds in Refill Solutions for Electronic Cigarettes Using HS-GCxIMS and Internal Standards

**DOI:** 10.3390/molecules27228067

**Published:** 2022-11-20

**Authors:** Alexander L. R. M. Augustini, Stefanie Sielemann, Ursula Telgheder

**Affiliations:** 1Hamm-Lippstadt University of Applied Sciences, Marker Allee 76-78, 59063 Hamm, Germany; 2Faculty of Chemistry, Instrumental Analytical Chemistry, University of Duisburg-Essen, Universitätsstraße 5, 45141 Essen, Germany

**Keywords:** ion mobility spectrometry, gas chromatography, headspace, flavor compounds, electronic cigarettes, e-liquids

## Abstract

New regulations on the use of flavor compounds in tobaccoless electronic cigarettes require comprehensive analyses. Gas chromatography coupled ion mobility spectrometry is on the rise as an analytical technique for analyzing volatile organic compounds as it combines sensitivity, selectivity, and easy usage with a full-range screening. A current challenge is the quantitative GCxIMS-analysis. Non-linear calibration methods are predominantly used. This work presents a new calibration method using linearization and its corresponding fit based on the relation between the reactant and analyte ions from the chemical ionization. The analysis of e-liquids is used to compare the presented calibration with an established method based on a non-linear Boltzmann fit. Since e-liquids contain matrix compounds that have been shown to influence the analyte signals, the use of internal standards is introduced to reduce these effects in GCxIMS-analysis directly. Different matrix mixtures were evaluated in the matrix-matched calibration to improve the quantitation further. The system’s detection and quantitation limits were determined using a separate linear calibration. A matrix-matched calibration series of 29 volatile compounds with 12 levels were used to determine the concentration of these substances in a spiked, flavorless e-liquid and a banana-flavored e-liquid, validating the quality of the different calibrations.

## 1. Introduction

Electronic cigarettes are a worldwide consumer product, with the regulations slowly catching up [1]. Analysis-wise, they present an entirely new product incomparable to other items in this area. Therefore, it is necessary to establish new protocols for simple and fast analysis. Flavor compound analysis is a widespread field that often employs gas chromatography–mass spectrometry (GC–MS) [2]. So many applications for electronic cigarettes and their refill solutions, e-liquids, also use this method [3,4,5]. Previously, regulators controlled very few but critical substances. A short list of concerning substances allows for a simple, targeted analysis. Currently, regulations are tightened, or further tightening is discussed. In some countries, banning a large spread of flavors or just allowing a fixed set of substances is debated [6,7,8].

With strengthening regulations, the demand for complex analysis methods increases as well. Primarily, this includes the analysis of flavor compounds. On top of the screening for ingredients, the reactivity or stability of e-liquid mixtures is still under investigation. Among others, the formation of acetals from flavor compounds in propylene glycol (PG) mixtures was observed. As these reaction products have an entirely different effect than their starting compounds, further screening targeting unknown substances is necessary [9]. Especially when extending this method to vapor analysis, a broad spectrum of unknown compounds can be expected [5,10,11].

Costly technical changes are necessary to improve the separation capabilities of a GC–MS. For example, a two-dimensional GCxGC or a more complex mass spectrometer can offer these capabilities. However, this makes the analysis more difficult and expensive. Therefore, we have established the use of a gas chromatograph coupled ion mobility spectrometer (GCxIMS) to allow a better separation, a more user-friendly analysis, and a more affordable setup [12,13]. In addition, transferring this analysis onto a portable onsite GCxIMS is also possible.

However, quantitative analysis based on GCxIMS is still challenging compared with more common GC-coupled detectors. Only a few methods have been discussed in depth [14,15,16]. There is no universally accepted procedure. The main concern is the small range in which the instrumental response can be approximated with a linear fit [17]. For this reason, quantitative analysis usually requires a non-linear approach [12,14].

The ionization method preferred for GCxIMS is the source of the small linear range. Using the radioactive atmospheric pressure chemical ionization (R-APCI), the ionization of analyte molecules is limited to the number of reactant ions available. Analyte molecules form protonated water clusters (monomers) during the ionization process. These can then be separated and detected in the drift tube. As the reactant ions are a limiting factor, they can be depleted at high analyte concentrations. This results in the additional formation of water clusters containing more than one analyte molecule (dimer) [18]. The included molecules can be identical or different, forming either dimers or mixed dimers (heteromers) [17,19]. The latter can even be energetically favorable [20]. The signals of all these different ionized clusters represent the amount of analyte present and must be considered to maximize the available range for quantitation. A graphical representation is shown in Figure 1 using the example of ethyl acetate in a simple e-liquid.

This work presents a new quantitation method for flavor compounds in e-liquids using the link between reaction and analyte ions to normalize the analyte signal, resulting in a linearized calibration and allowing a user-friendly method validation based on widespread procedures. We compare this new method with the manufacturer-recommended non-linear method based on a Boltzmann-type fit [14]. Both methods are evaluated using standard validation parameters, including goodness-of-fit, available quantitation range, and the recovery rate in a spiked e-liquid.

Data normalization is quite common in GCxIMS applications. As the ionization of the analytes is dependent upon the available reactant ions, the analyte ion peak (AIP) is often normalized to the reactant ion peak (RIP) [21,22,23]. However, these normalized values are then used in multivariate analysis, not for direct quantitation.

The normalization used for the linearization of the quantitative GCxIMS data is based on the law of mass action. It takes into account the reduction of available reactant ions as the concentration increases. Less available reactant ions reduce the frequency in which analyte molecules collide with unbound reactant ions, increasing the number of dimers formed and lowering the ionization efficiency overall. The relation of the various ions formed is shown for the example of ethyl acetate in Figure 1.

The matrix of e-liquids has been observed to influence the headspace during a quantitative analysis [24,25]. Therefore, the matrix effects have to be evaluated for their influence during any quantitation efforts. So, we evaluated different possible matrix compositions for their effect on quantitative results and chose an appropriate sample matrix for the necessary matrix-matched calibration.

Repeatability is a significant necessity for the application of quantitative analysis. Using internal standards is a common tool to reduce the fluctuations of the results due to the measurement system or the sample preparation. This tool is well-established for GC–MS methods. However, using an internal standard in GCxIMS analysis is quite rare. To our knowledge, only Zhu et al. have published fully quantitative results using an internal standard. In their work, it was used to monitor instrument performance and possibly correct for deviations of retention times [14]. This approach is used frequently in non-quantitative settings as well [23,26,27]. Therefore, our application of internal standards to improve the precision of quantitative analysis has not been published before.

To evaluate the capabilities of the GCxIMS, we describe a simple procedure to calculate the limits of detection and quantitation (LOD/LOQ) for this analysis. Based on the established DIN 32645:2008, we use a regular linear regression for this calculation. This calculation requires a narrow calibration that is close to the estimated LOD. The detector’s response for these low concentration levels can be approximated best with a linear fit and thus allows the application of the calibration method as described in the DIN 32645:2008. This procedure allows a more precise evaluation of the GCxIMS’ capabilities than the otherwise common estimations using the detector’s noise level. However, this calculation of the LOD and LOQ is separate from the calibration used in the quantitation. This has to be taken into account when evaluating the presented quantitation.

## 2. Results and Discussion

The investigations presented here for the calibration of the IMS for VOCs in e-liquids were carried out in a matrix similar to an e-liquid consisting of water, PG, and glycerol (GL). The influence of this matrix on the substances in the gaseous phase will be considered in detail in Section 2.2. For the quantitative observations, 29 substances were selected that represent a wide ratio of flavor compounds (see Table 1). These are substances of the functional classes of alcohols, terpenes, ketones, aldehydes, and esters. Although using certain substances in e-liquids, such as diketones, is already forbidden in some countries, other substances on this list are being considered for such restrictions.

The GCxIMS plot of a commercial banana-flavored e-liquid spiked with these substances at a mass concentration of 13 µg/g and the 7 2-alkanones as internal standards at 1 µg/g is shown in Figure 2. After dilution, the concentrations were 1.3 mg/L and 0.1 mg/L, respectively. The substances are separated in 35 min and unambiguously assigned retention and drift times. In addition to the spiked substances and the internal standards, 12 compounds were identified in the original e-liquid. These were preliminarily identified with the help of the parallel GC–MS using the strategy described in a previous publication [13]. Whenever possible, reference substances were used as verification. All identified substances are listed in Table 2.

An initial study determined that the substance concentration and signal intensity was represented best by using the maximum height of each peak. The utilized software tool can either express the total volume by integration or the maximum height above the lowest measured intensity in a predefined rectangular area. However, a sufficient baseline separation to use this method for a complex sample was not achievable within a reasonable measurement time. Therefore, the peak volume was too error-prone to be used. In this work, the signal intensity was calculated using the peak maximum over the baseline, as determined in the manually chosen rectangular areas. A graphical explanation of this situation is shown in the appendix in Figure A1.

Using the peak height requires similar conditions for calibration and sample measurements, as changes in the peak symmetry can distort the ratio of analyte to signal. These conditions were achieved by using the same chromatographic method and a matrix-matched calibration. When user-friendly software tools become available in the future that improve the peak-finding and integration of GCxIMS data, this topic should be revisited.

### 2.1. Repeatability and Internal Standards

A common problem when using an IMS as a detector is the ion suppression often observed when coelution occurs. Due to different proton affinities, the possible charge transfer between analytes or matrix components can lead to signal reduction. This effect makes quantitation difficult, sometimes even impossible [28,29]. In the case of e-liquids, large concentrations of PG and GL are present in the matrix. PG had to be taken into account, as it coeluted with certain investigated substances such as methyl-isobutyl ketone (MIBK) and 2,3-hexanedione (as seen in Figure 3). GL, however, was not detected by HS-GCxIMS and could therefore be ignored. The coelution with PG caused the formation of mixed dimers between the analyte and matrix molecules [30]. These were identified by observing the changes in the measurements when varying matrix and analyte concentrations independently. Mixed dimers were eliminated in measurements without PG and therefore are directly linked to this compound.

The use of internal standards was evaluated to correct signal suppression through the competing ionization of matrix molecules. The evaluation was performed by the described reproducibility test using 29 substances at four concentration levels (0.05, 0.3, 1, and 4 mg/L) in a flavorless e-liquid, with 11 repetitions. The concentration range was chosen to cover most of the possible dynamic range available with this setup. Unfortunately, for some compounds, the lowest (0.05 mg/L) and the highest (4 mg/L) concentration exceeded the dynamic range, resulting in very high standard deviations at these levels.

Internal standards are ideally chosen to have similar properties to the analytes, not to react with the sample, and not be present in the sample. The homologous series of 2-alkanones from 2-butanone to 2-decanone was selected, as they have similar properties compared with the investigated flavor compounds but do not have a high flavor value. They offer the additional advantage of being used as the base for the correlation with the parallel GC–MS via the identification strategy described in the corresponding publication.

Different effects on the analytes were compensated by choosing a range of substances as internal standards, covering the entire investigated retention time. Each analyte was compared with the two closest eluting 2-alkanones, choosing the substance with the most comparable signal intensity fluctuation as the internal standard. Using the ratio of the intensity of an analyte and the chosen internal standard reduced its variance significantly compared to the variance of the signal intensity of the analytes on their own. Figure 4 shows this for a selected set of compounds, covering the full range of retention times and compound classes. The results for all substances are shown in Figure A2, which is enclosed in Appendix A.

It is noticeable that the relative standard deviation shows the biggest improvements for most substances at lower concentrations (0.05 and 0.3 mg/L) when using the signal ratio of analyte and internal standards. These concentration levels are closer to the concentration used for the internal standard (0.1 mg/L). For substances at 0.05 mg/L, the standard deviation is reduced by an average of 5%, reducing the variance by at least one-fifth. The reduction at higher levels is comparable. However, since the total variance at these levels is far smaller, the improvements are not as extensive. The lowest concentration (0.05 mg/L) is close to the limit of quantitation. Thus, higher variances are expected and are reflected in the results in Figure 4. At the highest concentration (4 mg/L), the coefficient of variance increases slightly through the use of an internal standard for most substances. As the signal intensity at this concentration is very close to or sometimes even above the upper limit of the usable dynamic range, other effects have an influence than at the low concentrations for which the internal standards have been chosen.

The very high standard deviations for MIBK stand out. These deviations are due to the coelution of this compound with PG, a major matrix component that shows highly fluctuating signal intensities and huge signals. The slightly later eluting 2,3-hexanedione is less influenced. This substance also profits from the very closely eluting internal standard, 2-hexanone, compensating for matrix effects (see Figure 3).

The implementation of internal standards was shown to be especially useful in the same concentration range as the sample. The higher the difference of concentrations reaches, the more this advantage shrinks. Only furfural diverges from this rule. At low concentrations, the standard deviation increases by using 2-hexanone as an internal standard. Furfural has the highest LOD of all observed substances and shows, compared with 2-hexanone, only very low signal intensities at the same concentration level (compare Figure 5). Therefore, furfural needs to be used in higher concentrations compared to other substances to produce a similarly intense signal as the internal standard, explaining the exception from the previously mentioned rule. It might be more accurate to compare the ability of internal standards at the same signal intensity as the analyte. However, this is in routine analysis less practical. Nevertheless, a sensibly chosen internal standard can enable a successful approach to quantitation in such applications.

### 2.2. Matrix Effects

To study the influence of the matrix on the analysis and to identify a suitable matrix for the calibration, five different combinations of PG and GL were spiked with 5 µg/g flavor compounds and analyzed. The results for a selected set of substances compared to the same concentration of flavor compounds in water are shown in Figure 6. Again, Figure A3 showing all results is available in Appendix A.

Especially the unpolar, monoterpenic (+)-α-pinene and D(+)-limonene show a substantial effect, as the same concentration in the less polar matrices shows double the signal intensity as for the sample in water. As these compounds have only a very low solubility in water, the concentration of these compounds was probably lower in the aqueous than in the matrix-matched samples. This loss can be explained through evaporation or inhomogeneous distribution during dilution. More polar flavor compounds show a reduced signal intensity in the matrix. The prevalent state in this matrix is finding less than in the aqueous samples, as expected by the increasing solubility through GL and PG. In addition, the matrix-matched standard heavily affects substances coeluting with matrix components (MIBK, 2,3-hexanedione). It shows that adapting the calibration is necessary to adjust to the effects of the matrix on the analysis. For further investigations, it was decided to calibrate in a simple matrix of equal masses PG and GL (1 + 1), as it forms the average for the various combinations of e-liquids without sticking out in its effect on either the analysis or the consumer.

### 2.3. Quantitation

The non-linear correlation between signal intensity and concentration poses a major challenge when using the GCxIMS as a quantitative analyzer. Therefore, the choice of function to describe this correlation is very important.

In this analysis, the intensity of an analyte’s signal was taken as the maximum height of the signal above the background of this signal. Additionally, the analyte’s signal was divided by the signal intensity of an individually chosen internal standard to increase the repeatability. Since the internal standard’s concentration (0.1 mg/L) is identical in all calibrations and samples, it cancels out in the consequential calculations. Hence, it was left out of any plots.

The manufacturer recommends a Boltzmann-type regression to represent the relation between the analyte’s signal intensity and its concentration. As an example, this is shown for the three substances 2-methylpropanal (Figure 7A), furfural (Figure 5A), and menthol (Figure 8A). This function fits the curvature of the signal ratio and concentration relationship well, especially when considering the decreasing slope when approaching the maximum. It should be noted that even though the Boltzmann fit follows the curve at the saturation level, it is not possible to use a calibration at these levels. The signal intensity stays approximately constant when it approaches its maximum. Therefore, it is not possible to differentiate concentrations at this level and above. This is further described in Section 3.6.

As an alternative, we present a linearized approach. This method uses the signal intensity ratio of the ionized analyte and the remaining reactant ions (see Equation (Equation 5)) to compensate for the decreasing ionization efficiency of this detector at rising concentrations. The resulting correlation is best fitted linearly (see Figure 7B and Figure 8B). A linear fit simplifies the evaluation of the quantitation considerably. However, it was necessary to reduce the calibration range, as this method’s linearity is only applicable below the saturation level of the detector and well within the dynamic range. At high concentrations, the assumption that changes of ${\left[H_{2}O\right]}^{n-m}$ can be disregarded, as made in Section 3.7, loses validity. Areas outside the usable dynamic range are greyed out in the corresponding figures. For nearly all substances, the 5 mg/L level was outside this linearized range and needed to be rejected. This observation coincides well with the evaluation of the dynamic range without the linearization as described in Section 3.6.

The plotted validation samples fit the calibrations with a recovery rate spanning 66–104% (Boltzmann fit) and 58–108% (linearized fit) for the 1.0 mg/L samples with a prevalence for lower-than-expected findings. This is expanded on in Figure 9 and further discussed in Section 2.4.

The full range of the possible calibration (approx. 0.01–5 mg/L) was intentionally chosen to be larger than what is accepted as good practice to evaluate the capabilities of the quantitation. While adjusting the fits used for the calibration, the range was adapted to a usable size for each substance individually. The lower limit of the range for the quantitation is restricted by the method’s LOQ as was separately determined (see Section 3.5) spanning 7–60 µg/L. However, both types of approximation do not necessarily fit the lower concentrations for all substances well, without accepting large deviations for the higher concentrations and thus restricting the usable range further. The same is to be said about the upper limit. Whereas the Boltzmann fit describes the changing response pretty well, it is not perfect in following the entire span. The linearization, as already mentioned, requires certain assumptions that lose validity when the reactant ions become drained. Thus, the lower limit spans 0.01–0.11 mg/L for both calibrations. The upper limit spans 0.88–5.58 mg/L for the Boltzmann fit and 0.63–5.54 mg/L for the linearized fit. Even though the limits are different for the individual substances, the Boltzmann fit covers the larger span for most substances, while the average RE is equal for both calibrations. Future endeavors using implicit functions may better approximate the relationship between signal intensity and concentration. However, this is beyond the scope of this work.

The listed ranges in Table 1 reflect these limitations, as optimized for the best fit. Especially, substances that suffer from many coeluting substances, such as 2-methylpropanal and diacetyl, tend to coelute with the partially tailing, non-retarded, unidentified contaminations. Isoamyl alcohol, MIBK, 2,3-hexanedione, and hexanal are heavily affected by the coeluting PG and thus were calibrated for a small concentration span as well. The monomer of *trans*-2-hexenol has a very short drift time and thus partially overlaps with the RIP.

### 2.4. Recovery and Application

The recovery was evaluated with a spiked flavorless, commercial e-liquid. As shown in Figure 9, the recovery rate for various substances shows widely varying results, with a prevalence of lower-than-expected findings. MIBK and 2,3-hexanedione, known to coelute with PG, show strongly deviating results. This was expected from the varying intensity of the PG signal in these measurements. Large deviations are also noticeable for the early eluting substances. In this case, a coeluting contaminant is a plausible reason influencing the background level. D(+)-limonene and α-pinene show relatively low recovery rates. As these two have shown differing effects while examining the matrix as well, this might be an effect of their very low polarity and the resulting high volatility.

Apart from this, the lower concentrations show lower recoveries than the higher ones. This is a common effect for measurements in a complex matrix, as any matrix shows a comparatively lower influence on higher analyte concentrations. Nevertheless, the recovery rates are comparable with the results published by Zhu et al. for flavor compounds in a wine matrix. The results for all evaluated substances are shown in Figure A4 in Appendix A.

As a demonstration of the applicability of this method, a banana-flavored e-liquid was analyzed, and the identified flavor components were quantified using the available calibrations. The results were then converted to reflect the amount of each substance in the undiluted e-liquid and are listed in Table 3. Hexanal, butyl acetate, and D(+)-limonene, identified in the e-liquid as well, produced signals above the detector’s dynamic range. Therefore they could not be quantified.

The differences between the two calibration methods are minor. There are three noteworthy disagreements. The deviations of the calculated content of menthol can be explained by its concentration, which is close (linear fit) and partially above (Boltzmann fit) the upper limit of the calibration. At this point, the uncertainty increases drastically, which also comes up in the standard deviation. The ethyl acetate content shows a slight difference and the octanal content shows a larger difference for the two calibration methods. However, this might be explained by the relatively large deviations of the calibration points from the ideal fit, as expressed in the relative error (RE) and root mean squared error (RMSE) (see Table 1). Improving the calibration would correct this discrepancy. Additionally, ethyl acetate coelutes with ethanol, which was not included in the matrix-matched calibration.

The flavored e-liquid shows a strong ethanol signal (as displayed in Figure 2), which is known to be a hindrance in quantitative analysis by GCxIMS [14]. Due to the low proton affinity of ethanol compared to other flavor compounds, for example, ethyl acetate [31], a quantitative analysis is still possible. An in-depth evaluation of the influence of ethanol in an e-liquid, comparable to the work of Zhu et al., can help evaluate this problem when a high ethanol content is a common occurrence.

## 3. Materials and Methods

### 3.1. Chemicals and Samples

Reference substances were purchased with sufficient purity (90% or more) from Fisher Scientific (Schwerte, Germany), Carl Roth (Karlsruhe, Germany), TCI Deutschland (Eschborn, Germany), and Sigma-Aldrich (Taufkirchen, Germany). More details are given in Appendix A Table A1. 1,2-propanediol (PG) and 1,2,3-propanetriol (GL), water-free, were both purchased from Carl Roth at least 99.5% pure. The water used in this research was ultra-pure water, produced with a Veolia Elga Purelab flex 4 (Celle, Germany). Helium 5.0 (purity at least 99.999%) as separation gas was purchased from Messer Industriegase (Siegen, Germany). Nitrogen was supplied by an in-house generator with a purity of at least 99.999% and filtered with a hydrocarbon trap (Supelpure HC from Supelco, Bellefonte, PA, USA) to increase its purity and protect against contamination.

Before the measurement, e-liquids were stored at room temperature and diluted 1 to 9 parts by mass with ultra-pure water. Stock solutions were prepared in PG, but for the preparation of working solutions, these were further diluted in water. All reference standards were treated as if neat, except the separated enantiomers L-Menthone, D-Menthone, Neral, and Geranial. These were weighed in together, as if neat, however then using the results from an external analysis, were corrected using a factor describing the ratio in which they were present. This procedure resulted in lower concentrations for each of these separate substances. Stock solutions were stored for short periods of time at 4 °C. Whenever they needed to be stored for two or more weeks, they were kept at −16 °C.

The internal standard solution was prepared by diluting equal masses of 2-butanone, 2-pentanone, 2-hexanone, 2-heptanone, 2-octanone, 2-nonanone, and 2-decanone in water to achieve a final concentration of 1 mg/L and stored at −16 °C. This solution was added to all samples or solutions before the final dilution step, resulting in a concentration of 0.1 mg/L in the analyzed solution.

The matrix-matched calibration was prepared in a mixture of equal parts PG and GL. This and other mixtures were used to evaluate the matrix effects.

For the recovery and validation study, a flavorless e-liquid (Vape-Base, VapeBase GmbH, Essen, Germany), consisting of 55% PG, 35% GL, and 10% water, containing 3 mg/g nicotine, was spiked with the appropriate amount of the reference substances. The analysis method was tested on a nicotine-free e-liquid from a polish retailer with a fruity banana flavor.

### 3.2. Sample Preparation and Analysis

All samples and reference solutions were mixed with an equal volume of internal standard solution (1 mg/L) and topped off with water to tenfold the initial volume. Indications of an analyte content are given as weight per weight matrix before this final dilution. However, when a concentration is mentioned, this includes this final dilution, which is calculated in weight per volume. For the incubation in the autosampler (30 min. at 120 °C), 1 mL of the diluted solution was filled into a 20 mL crimp-closed headspace vials (polytetrafluoroethylene/polysiloxane septa, CS-Chromatographie Service, Langerwehe, Germany); 2 mL of the sample’s headspace was injected at a split of 1:10 into the GC. The analysis was performed on a HS-GCxIMS/GC–MS system based on a two flow line Shimadzu GC-2010 (Kyoto, Japan) on HP-5 MS UI (Agilent, Santa Clara, CA, USA) 30 m × 0.25 mm × 0.5 µm columns, running a temperature gradient at a helium flow rate of 35 cm/s as described in a previous publication [13]. IMS measurements were performed on a manufacturer-improved IMS device (G.A.S. mbH, Dortmund, Germany) with a drift tube of 15.2 × 53 mm, ionization by ^3^H-source, at 80 °C, a nitrogen drift gas flow rate of 150 mL/min, a field strength of 500 V/cm and a repetition rate of 40 Hz. The detector noise was reduced by averaging each data point from twelve measurement events, resulting in a data collection rate of about 3 Hz, using the RC 1.02 IMS OEM Module software by G.A.S.mbH.

### 3.3. Data Processing and Interpretation

The collected results of the GCxIMS were plotted as a heat map, as shown in Figure 2 to express the three-dimensional data. The drift time of the separation in the drift tube IMS is plotted on the x-axis. The retention time of the GC separation is plotted along the y-axis. The z-axis or color scale represents the intensity as detected. Additionally, the data were processed with the LAV software suite (version 2.2.1, G.A.S. mbH) using a Savitzky–Golay-filter.

Substances were identified by their peak position with their specific drift and retention time. The drift times were normalized to the RIP position. Unknown substances were tentatively identified using the previously published method using a GC–MS [13]. The identification was validated using reference substances whenever possible. The difference between the highest and lowest data point within a predefined area was taken for each substance using the “Height above Area Minimum”-ability of the VoCal Software (version 1.3, G.A.S.mbH) as the signal intensity.

The data were managed and organized using Excel 2019 (Microsoft, Redmond, WA, USA), including simple calculations. Finally, plots were created and fitted using OriginPro 2022 (OriginLab, Northampton, MA, USA). Whenever multiple measurements are shown, the result is presented as the arithmetic mean using the standard deviation as error bars.

### 3.4. Outlier

It was noticed that nearly every series of measurements included one or more samples with highly deviating results. Closer inspection showed very large signals for PG in these measurements. Due to the ionization by R-APCI, an abundance of matrix molecules in the ionization chamber can suppress the ionization of analyte molecules. However, this would only explain the suppression of coeluting substances. Analytes that elute before the matrix are affected as well. Therefore the ionization is not a sufficient explanation. Our current theory is based on the formation of droplets on the underside of the vial’s septum, whenever a prepared vial was stored for a couple of hours before the measurement. This is inevitable as freshly preparing every sample immediately before the analysis is impractical. The effect was observed independently of the temperature at which the samples were stored (4 °C or room temperature). Whenever a sample is drawn and the needle touches such a droplet, it is contaminated with this liquid, drastically changing the injected sample’s composition. Since the measured effect is very large and affects the internal standards as well, these outliers are easily noticeable and simple to remove or repeat the measurement. However, it requires every measurement to be at least repeated or even measured in triplicate.

### 3.5. Limits of Detection and Quantitation

The main validation study determined the LOD and LOQ separately with five repetitions of a seven-point calibration series (0.02–2 µg/g matrix, corresponds to 2–200 µg/L diluted sample). Due to the very low analyte concentration compared to the available reactant ions, this calibration range is best approximated using a linear fit. This calibration, which is very close to the approximated LOD, allows the “calibration straight” method for calculating the LOD and LOQ as described in DIN 32645:2008. The uncertainty of the blank values is calculated using the calibration data instead of the direct measurement of blank samples. This circumvents problems with non-normally distributed blank values and a heteroscedastic variance. Equation (Equation 1) was used for these calculations, with the residual standard deviation ($s_{x,y}$) over the slope (*b*), the quantile of the t-distribution (*t*) for n−2 degrees of freedom (*f*) and α-error (here: 0.05), number of measurement repetitions (*m*), number of calibration measurements (*n*), average value of the calibration ($\bar{x}$), and the sum of the deviations squares ($Q_{x}$). The LOQ was calculated with Equation (Equation 2), using the two-sided *t*-test and k=3.
(1)$$\begin{array}{c} x_{LOD}=\frac{s_{x,y}}{b}\cdot t_{f;\alpha }\cdot \sqrt{\frac{1}{m}+\frac{1}{n}+\frac{{\bar{x}}^{2}}{Q_{x}}} \end{array}$$
(2)$$\begin{array}{c} x_{LOQ}=k\cdot \frac{s_{x,y}}{b}\cdot t_{f;\frac{\alpha }{2}}\cdot \sqrt{\frac{1}{m}+\frac{1}{n}+\frac{{\left(k\cdot x_{LOD}-\bar{x}\right)}^{2}}{Q_{x}}} \end{array}$$

### 3.6. Dynamic Range

R-APCI-based IMS is known to have a limited dynamic range [14]. The LOD and LOQ set the lower limits. These were determined using calibration within the linear dynamic range at the approximated LOD as described in Section 3.5. The upper limit of this application is determined as the lowest calibration level at which the signal intensity is no longer sufficiently different (*p*-value, p>0.1) from the highest or second-highest calibration level. This was proven using the independent two-sample *t*-test assuming heteroscedasticity, as first described by Welch [32]. Heteroscedasticity or non-equal variance is given as the variance increases unproportionally at the upper level of the dynamic range.

### 3.7. Quantitation Method

Quantitation in GCxIMS applications using R-APCI requires non-linear approximations to cover more extensive concentration ranges. The ionization of analytes is limited by the available reactant ions, thus limiting the dynamic range and the resulting possible linear dynamic range even further. However, the change of ionization efficiency at different analyte concentrations varies only through the fewer available reactant ions. So normalizing the signal intensity to the available reactant ions negates this effect and results in a nearly linear relation between analyte signal and analyte concentration. This normalization is derived from the law of mass action, describing the ionization reaction of analyte molecules (A) using the reactant ions (H^+^(H_2_O)_*n*_) to create the ionized analyte water clusters ([AH^+^(H_2_O)_*m*_]).

For the simplification, as shown in the Equation (Equation 3) it is assumed that at equilibrium state [AH^+^(H_2_O)_*m*_] is proportional to the intensity of the AIP and [H^+^(H_2_O)_*n*_] is proportional to the RIP which is equal to the initial RIP (RIP_0_) minus the AIP. Furthermore, it assumed that only a tiny fraction of the available analyte molecules are ionized [18] (p. 180). Therefore, [A] is approximately equal to [A_0_]. The basic principle of this has already been considered by Achim Schumann, 2001 [33].
$$\begin{array}{c} A+H^{+}{\left(H_{2}O\right)}_{n}\;⇄\;{AH}^{+}{\left(H_{2}O\right)}_{m}+\left(n-m\right)H_{2}O \end{array}$$
(3)$$\begin{array}{cc} K & =\frac{\left[AH^{+}{\left(H_{2}O\right)}_{m}\right]\cdot {\left[H_{2}O\right]}^{n-m}}{\left[A\right]\cdot [H^{+}{\left(H_{2}O\right)}_{n}]}\\ K & =\frac{AIP\cdot {\left[H_{2}O\right]}^{n-m}}{\left[A_{0}\right]\cdot \left(RIP_{0}-AIP\right)}\;\mathrm{for}\left[AH^{+}{\left(H_{2}O\right)}_{m}\right]\propto AIP\;\mathrm{and}\;\left[A\right]\approx \left[A_{0}\right] \end{array}$$
(4)$$\begin{array}{cc} & \mathrm{Converted}\mathrm{into}a\mathrm{linear}\mathrm{equation}:\\ & \frac{K}{{\left[H_{2}O\right]}^{n-m}}\cdot \left[A_{0}\right]=\underset{\mathrm{normalized}\;\mathrm{signal}\;\mathrm{intensity}}{\underset{︸}{\frac{AIP}{RIP_{0}-AIP}}} \end{array}$$

However, the formation of dimer ions needs to be considered, where applicable (see reaction in Equation (Equation 5)). It is not possible to separate both reactions as the dimers are formed from the monomers. Therefore, the measurable amount of monomers has been reduced by the amount of dimers. The theoretical amount of monomers (in Equation (Equation 5) marked with *), that would have been formed without any dimerization can be approximated by using the sum of the different detected analyte clusters.
$$\begin{array}{c} A+AH^{+}{\left(H_{2}O\right)}_{m}\;⇄\;A_{2}H^{+}{\left(H_{2}O\right)}_{m} \end{array}$$
(5)$$\begin{array}{cc} & \frac{K^{*}}{{\left[H_{2}O\right]}^{n-m}}\cdot \left[A_{0}\right]=\frac{AIP^{*}}{RIP_{0}-AIP^{*}}\approx \frac{AIP_{M}+AIP_{D}}{RIP_{0}-\left(AIP_{M}+AIP_{D}\right)} \end{array}$$

As an example, a fit of the linked monomer and dimer signal intensities, as well as their sum, the corresponding RIP, and the linearized function are shown in Figure 1 for the matrix-matched calibration of ethyl acetate spanning 0.02–5 mg/L.

As comparison we used the commonly applied Boltzmann-type regression (see Equation (Equation 6)), as already shown in Figure 1. This has most recently and in-depth been discussed by Zhu et al. [14]. The sum of the intensities of monomer, dimer, and any relevant mixed dimer signals were used to approximate the relationship between the analyte’s concentration and the corresponding signal
(6)$$y=b+\frac{a-b}{1+e^{\left(ln(x)-c)/d\right)}}$$

The entire range between the limit of detection and the upper limit of the dynamic range (as described in Section 3.6) was evaluated for the quantitation ranges using a calibration series for all substances spanning about 0.02–5 mg/L (undiluted: 0.2–50 µg/g matrix). However, neither the Boltzmann fit nor the linearization could cover the entire range reliably for all investigated substances. Thus, different ranges for the fits were compared before choosing the best fit by optimizing for low residuals and a suitable concentration range.

The goodness of fit was compared by calculating the concentrations of the analytes using the linearized and the Boltzmann regression from the calibration. The RMSE (see Equation (Equation 7)) and RE (Equation 8) were used as parameters [14,16].
(7)$$RMSE=\sqrt{\frac{\sum_{i=1}^{n}{\left(c_{i}-c_{i}^{calc}\right)}^{2}}{n}}$$
(8)$$RE=\sqrt{\frac{\sum_{i=1}^{n}{\left(c_{i}-c_{i}^{calc}\right)}^{2}}{\sum_{i=1}^{n}{\left(c_{i}\right)}^{2}}}$$

### 3.8. Evaluation of the Matrix

For the evaluation of the matrix effects, five different matrix compositions were compared. Pure water, pure PG, pure GL, and mixtures of PG and GL 1 + 4, 4 + 1, and 1 + 1 were spiked with 5 µg of the analyte mixture per gram matrix, diluted, and analyzed in quadruplicate, as described.

### 3.9. Validation, Recovery, and Application

The method’s validity was checked by spiking a flavorless, commercial e-liquid (Vape-Base) with four different concentrations (about 0.5, 3, 10, and 40 µg/g) of the selected 29 substances. These validation samples were mixed with the internal standard solution and diluted with water, just like the calibration samples. Eleven measurement runs of each concentration level were evaluated. The method’s applicability was demonstrated by performing a fourfold measurement of a banana-flavored e-liquid. The identified substances were matched with the available calibrations, and the analyte content was quantified.

## 4. Conclusions and Outlook

Calibration guarantees the trueness of analytical results. This work shows a method for practicable quantitation using GCxIMS by analyzing flavor compounds in e-liquids. It improves on previous shortcomings in this field. Especially, the available low LOD and LOQ and the high separation power make the GCxIMS a valuable tool without the need for more expensive and complex setups.

The limited linear dynamic range of the R-APCI is still considered a challenge. The recommended Boltzmann fit and the newly proposed linearization method were applied to obtain a feasible calibration function. The comparison shows the advantages of both methods. The fitted regressions show similar deviations for measured results as described by the RMSE and RE, which allow the assessment of the goodness of fit.

The Boltzmann model can cover slightly larger concentration ranges, as it can include results up to the saturation limit of the detector. However, the unique feature of this non-linear fit—to approach the maximum with a lower slope—comes at the cost of an increasing error at these high concentrations.

The model for linearized quantitation based on the normalized signal intensity offers much simpler calculations, as most users are accustomed to linear fits. The normalization is a simple process and can be automated, as only the initial RIP has to be recorded additionally. In this work, the initial RIP was determined during the recorded dead time before the elution of the first analytes and did not necessitate any further measurements.

The normalized signal intensity requires certain assumptions to be calculated. These assumptions lose validity at high concentrations near the upper limit of the detector’s dynamic range. This effect reduces the applicable range of the linearization compared to the non-linear fit. However, neither fit could cover the entire evaluated dynamic range of the detector, thus requiring the fit to be adapted to the task.

Another common obstacle in GCxIMS analysis is the influence of the matrix through competing ionization whenever coelution occurs. Internal standards were used to compensate for these effects and other negative influences on the repeatability. The implementation was especially useful in the same signal intensity range as the sample. Therefore, adjusting not solely by concentration but also by response is necessary.

In addition to the approach presented here, other ionization techniques can reduce competing ionization effects. This includes the use of dopants [34], another ionization source [35], or the application of a High Kinetic Energy IMS (HiKE-IMS) [36]. Unfortunately, these either reduce the scope of possible analytes or are not yet commercially available. Using multivariate regression, Brendel et al. have presented an alternative method for quantifying complex mixtures using GCxIMS. This method has shown promise for complex problems. However, it changes the quantitation procedures drastically and is not yet favored in routine applications due to its complexity [16].

In summary, quantification with GCxIMS is feasible. Non-linear or normalized calibration models should be used to obtain the best possible results. A linear approximation can be used as a simple solution for small concentration spans. This allows, for example, a fast estimation of LOD and LOQ. Common procedures, for example, internal standards, are proven to be useful. Even though competing ionization due to coelution cannot be compensated in full by them, these options expand the scope of applications for the GCxIMS-technology. Alongside the already increasing interest in the non-targeted analysis, this also allows the quantitation of marker compounds, which play an important role in quality control or medical applications.

## Figures and Tables

**Figure 1 molecules-27-08067-f001:**
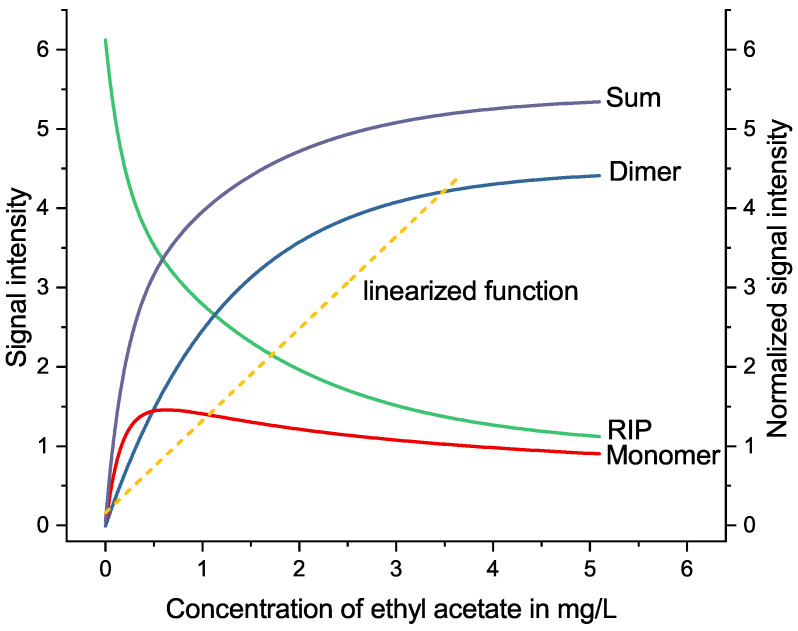
Fitted plot showing the analyte signal intensities for monomer and dimer, as well as the RIP for an increasing concentration of ethyl acetate in a sample in relation to the linearized function.

**Figure 2 molecules-27-08067-f002:**
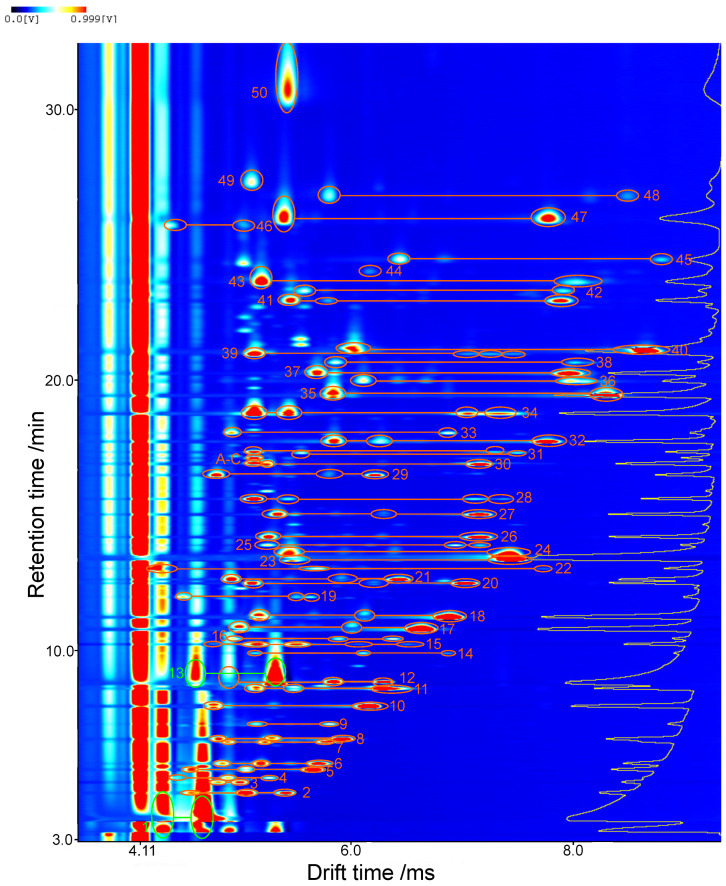
Partial GCxIMS plot of a banana-flavored nicotine-free e-liquid spiked with approximately 13 µg/g of 29 different flavor compounds. The corresponding monomer, dimer, and mixed dimer signals of all identified substances are labeled and connected with the substance names listed in Table 2.

**Figure 3 molecules-27-08067-f003:**
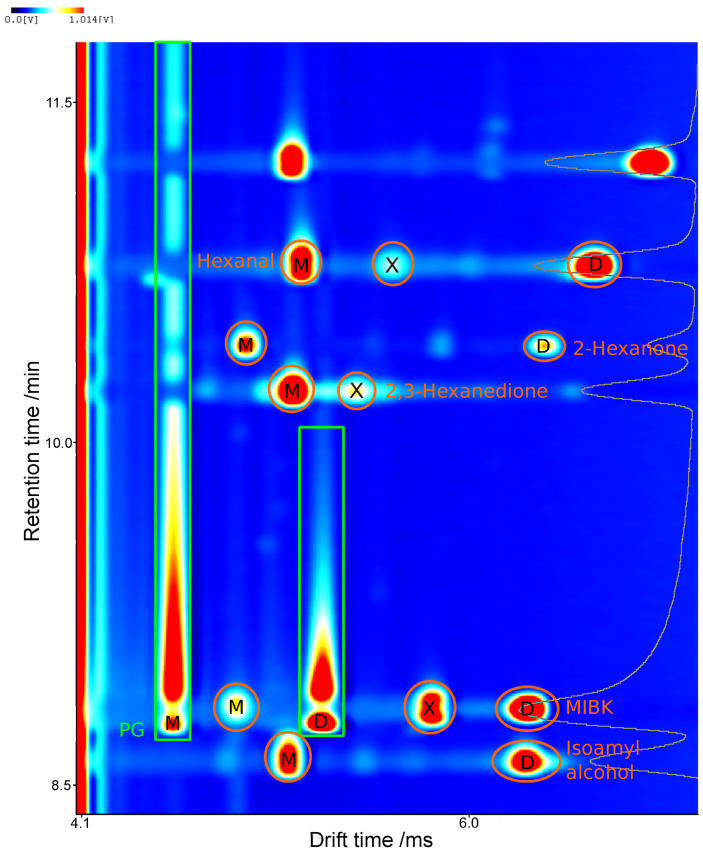
Section of a spiked, otherwise flavorless e-liquid, showing the coelution of propylene glycol, MIBK, 2,3-hexanedione, and the resulting creation of mixed dimers (X) between the drift times of monomers (M) and dimers (D).

**Figure 4 molecules-27-08067-f004:**
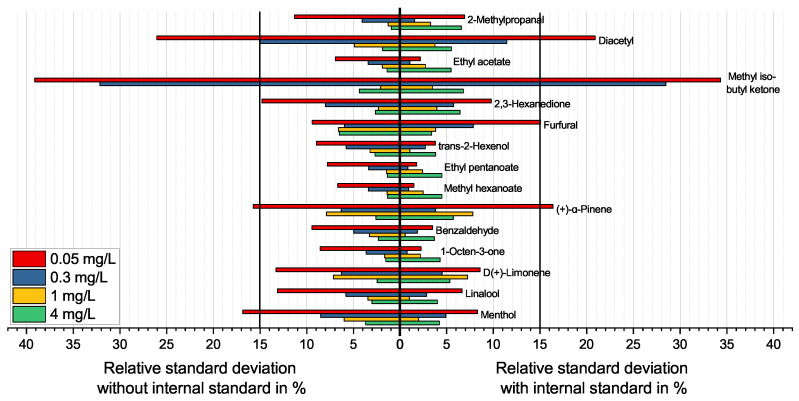
Diagram comparing the coefficient of variance or relative standard deviation of the signal intensity for selected standard substances in a flavorless e-liquid at four different concentration levels and eleven measurements each, with or without the use of an internal standard to reduce matrix effects.

**Figure 5 molecules-27-08067-f005:**
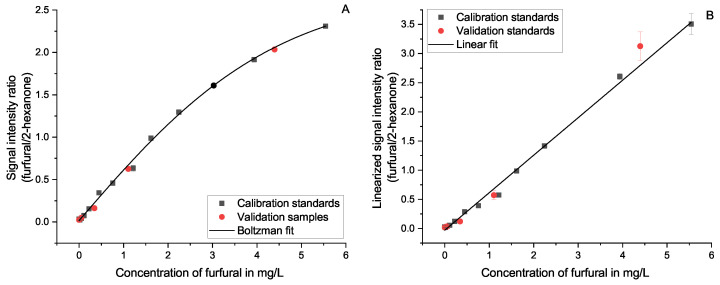
Calibration regression for furfural using 0.1 mg/L 2-butanone as an internal standard in matrix; based on a Boltzmann function (**A**) and a linearly converted function (**B**).

**Figure 6 molecules-27-08067-f006:**
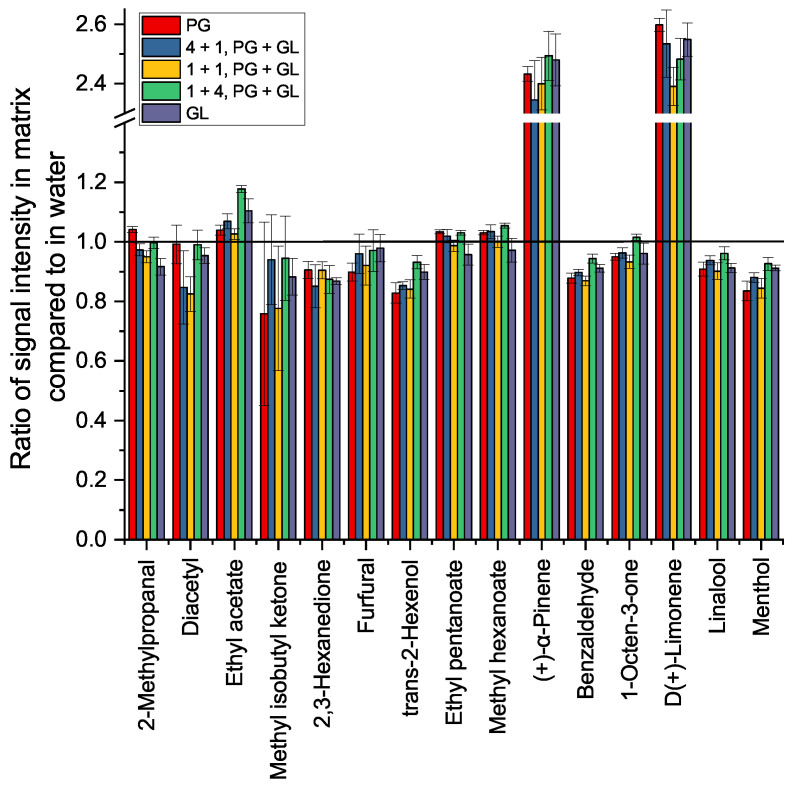
The signal intensity ratio for selected analytes in a range of possible matrices consisting of propylene glycol and glycerol, compared to in water for spiked samples of 5 µg/g matrix and quadruplicate measurements.

**Figure 7 molecules-27-08067-f007:**
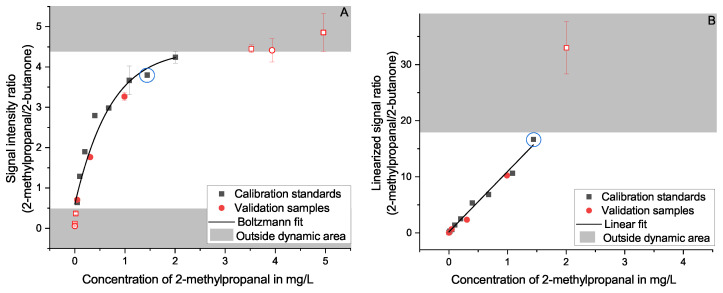
Calibration regression for 2-methylpropanal using 0.1 mg/L 2-butanone as an internal standard in matrix; based on a Boltzmann function (**A**) and a linearly converted function (**B**), showing the usable dynamic range of both regressions. The upper limit of the linearization (1.44 mg/L) is marked with a blue circle in both plots to clarify the different usable dynamic ranges. The area outside the usable dynamic range is colored gray, and the data points are colored light red and are unfilled to illustrate that they are not included in the calculated regression.

**Figure 8 molecules-27-08067-f008:**
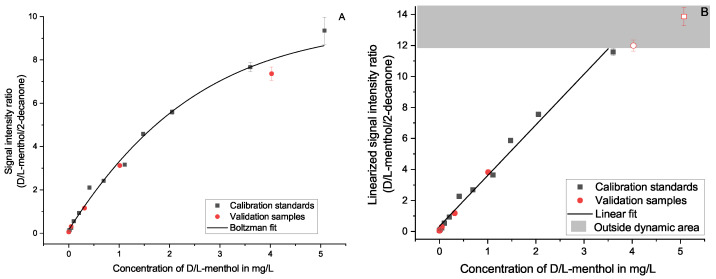
Calibration regression for Menthol (D- and L-isomers) using 0.1 mg/L 2-decanone as an internal standard in matrix; based on a Boltzmann function (**A**) and a linearly converted function (**B**). The area outside the usable dynamic range is colored gray, and the data points are colored light red and are unfilled to illustrate that they are not included in the calculated regression.

**Figure 9 molecules-27-08067-f009:**
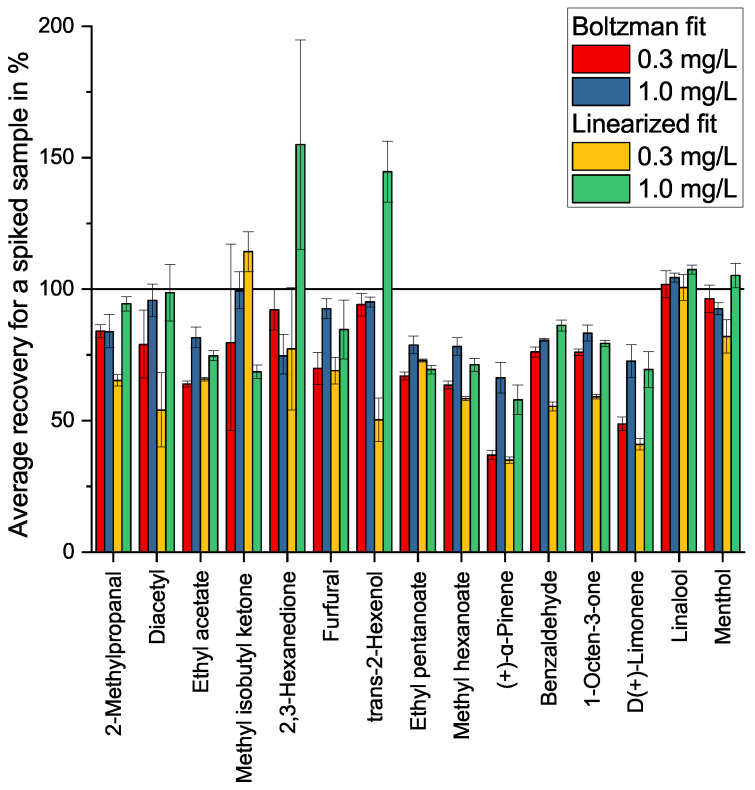
Recovery, as calculated using two different regressions for the calibration, for two different spiked concentrations and a selected set of substances in a flavorless e-liquid, with triplicate measurements.

**Table 1 molecules-27-08067-t001:** Summary of the results for the analyzed substances used in the method validation, sorted by their retention time (Rt), including the reduced mobility ($K_{0}$) for the monomer (^M^), mixed dimer (^X^), and dimer (^D^) signals, the limit of detection and quantitation (LOD and LOQ), the concentration range that was used for the Boltzmann fit and the linearized fit, including the root mean squared error (RMSE), the relative error (RE), and the coefficient of determination (r^2^).

Substance ^internal standard^	Rt/min.	$K_{0}^{M}$/Vs/cm^2^	$K_{0}^{X}$/Vs/cm^2^	$K_{0}^{D}$/Vs/cm^2^	LOD/µg/L	LOQ/µg/L	Boltzmann Fit			Linearized			
							Range/mg/L	RMSE/mg/L	RE/%	Range/mg/L	RMSE/mg/L	RE/%	r^2^
2-Methylpropanal ^C4^	4.8	1.824	1.655	1.538	11.1	38.0	0.05–2.01	0.11	10.95	0.02–1.44	0.06	8.86	0.99
Diacetyl ^C4^	5.1	1.732	-	-	6.7	23.9	0.11–3.91	0.09	5.06	0.11–2.22	0.10	10.29	0.98
Ethyl acetate ^C4^	5.6	1.820	-	1.468	5.0	18.1	0.10–3.62	0.10	5.87	0.05–2.06	0.11	10.40	0.98
Isobutanol ^C4^	5.8	1.717	1.599/1.525	1.456	9.3	32.2	0.11–3.75	0.09	5.29	0.11–3.75	0.17	11.72	0.98
2-Methylbutanal ^C4^	6.7	1.730	1.569	1.404	14.7	49.6	0.10–1.99	0.13	12.21	0.02–1.99	0.10	10.32	0.98
Isoamyl alcohol ^C5^	8.6	1.615	1.565	1.320	11.1	38.1	0.10–2.04	0.10	9.28	0.10–2.04	0.11	12.14	0.97
Methyl isobutyl ketone ^C5^	8.8	1.695	1.424	1.320	5.8	20.6	0.10–3.52	0.23	14.34	0.05–2.01	0.17	19.52	0.94
2,3-Hexanedione ^C6^	10.2	1.617	1.520	-	10.0	35.0	0.02–1.23	0.11	18.32	0.11–1.64	0.11	14.34	0.96
Hexanal ^C7^	10.8	1.595	1.473	1.256	13.5	45.8	0.11–3.84	0.20	11.67	0.11–3.84	0.08	7.84	0.99
Butyl acetate ^C6^	11.2	1.609	1.358	1.208	14.7	49.8	0.05–2.16	0.11	9.98	0.02–3.79	0.09	5.84	0.99
Furfural ^C6^	12.0	1.852	-	1.476	17.4	59.7	0.02–5.54	0.07	3.01	0.11–5.54	0.10	4.83	1.00
Ethyl 2-methyl butanoate ^C6^	12.5	1.628	-	1.182	16.3	55.0	0.10–3.57	0.29	17.89	0.10–3.57	0.07	4.58	1.00
*trans*-2-Hexenal ^C6^	12.6	1.688	1.412	1.290	13.4	45.8	0.05–2.05	0.09	9.25	0.05–5.07	0.12	6.12	0.99
*trans*-2-Hexen-1-ol ^C7^	13.0	1.956	-	1.460	4.4	15.9	0.05–3.72	0.15	9.74	0.10–2.12	0.12	13.14	0.97
Ethyl pentanoate ^C7^	14.2	1.583	-	1.162	16.0	54.1	0.10–5.05	0.28	12.21	0.05–5.05	0.08	8.71	0.99
Methyl hexanoate ^C8^	15.0	1.558	-	1.162	14.4	49.0	0.10–5.11	0.28	12.34	0.10–3.63	0.08	5.03	0.99
(+)-α-Pinene ^C8^	15.6	1.622	1.529/1.172	1.127	11.4	39.2	0.02–5.28	0.06	2.64	0.02–5.28	0.14	6.94	0.99
Benzaldehyde ^C7^	16.5	1.736	-	1.337	7.2	25.7	0.05–3.72	0.07	4.63	0.10–3.72	0.07	4.66	1.00
1-Octen-3-one ^C8^	16.9	1.590	1.530/1.235	1.161	9.9	34.3	0.10–5.16	0.24	10.40	0.10–5.16	0.09	4.45	1.00
Octanal ^C8^	17.7	1.422	1.325	1.071	13.9	47.1	0.10–3.51	0.29	18.35	0.10–3.51	0.09	9.50	0.98
*trans*,*trans*-2,4-Heptadienal ^C8^	18.0	1.690	-	1.211	9.7	33.9	0.02–5.58	0.07	2.91	0.02–5.58	0.17	7.39	0.99
D(+)-Limonene ^C8^	18.8	1.622	1.529	1.180/1.136	7.0	25.0	0.02–3.41	0.05	3.64	0.02–4.81	0.05	4.16	1.00
1-Octanol ^C9^	19.9	1.362	-	1.043	3.4	12.5	0.05–3.72	0.08	5.05	0.10–2.12	0.11	11.81	0.98
Linalool ^C9^	21.0	1.621	-	1.182/1.147/1.111	6.5	23.1	0.10–5.06	0.23	10.09	0.10–2.05	0.11	12.10	0.97
L-Menthone ^C9^	22.9	1.528	-	1.056	4.3	15.5	0.09–4.30	0.19	9.79	0.09–3.05	0.10	8.81	0.99
D-Menthone ^C9^	23.3	1.475	-	1.039	2.1	7.2	0.01–0.88	0.02	4.16	0.02–0.63	0.01	7.11	0.99
DL-Menthol ^C10^	23.6	1.600	-	1.041	3.2	11.9	0.02–3.60	0.08	5.45	0.10–3.60	0.08	8.77	0.99
Neral ^C9^	25.8	1.652	-	1.515	7.3	24.8	0.02–2.36	0.04	4.02	0.05–0.95	0.03	8.15	0.99
Geranial ^C9^	26.8	1.419	-	0.969	9.8	33.0	0.01–2.77	0.04	3.68	0.03–1.97	0.05	6.44	0.99

^C4–C10^ Internal standard that was used for this substance: 2-butanone (C4), 2-pentanone (C5), 2-hexanone (C6), 2-heptanone (C7), 2-octanone (C8), 2-nonanone (C9), 2-decanone (C10).

**Table 2 molecules-27-08067-t002:** List of substances that were identified in the partial GCxIMS plot of a banana-flavored and spiked e-liquid, shown in Figure 2. Substances that were added are marked ^add^, substances used as internal standards ^istd^, substances tentatively identified using the GC–MS ^MS^ and afterward verified using a reference standard ^ref^. Matrix compounds are marked ^matrix^.

#	Substance
1	Ethanol ^matrix^
2	2-Methylpropanal ^add^
3	Diacetyl ^add^
4	2-Butanone ^istd^
5	Ethyl acetate ^add^
6	Isobutanol ^add^
7	2-Methylbutanal ^add^
8	3-Methylbutanal ^ref^
9	2-Pentanone ^istd^
10	Ethyl propanoate ^ref^
11	Isoamyl alcohol ^add^
12	Methyl isobutyl ketone ^add^
13	Propylene glycol ^matrix^
14	Isobutyl acetate ^ref^
15	2,3-Hexanedione ^add^
16	2-Hexanone ^istd^
17	Hexanal ^add^
18	Butyl acetate ^add^
19	Furfural ^add^
20	Ethyl 2-methylbutanoate ^MS^
21	*trans*-2-Hexenal ^add^
22	*trans*-2-Hexenol ^add^
23	3-Methylbutyl acetate ^ref^
24	2-Methylbutyl acetate ^MS^
25	2-Heptanone ^istd^
26	Ethyl pentanoate ^add^
27	Methyl hexanoate ^add^
28	(+)-α-Pinene ^add^
29	Benzaldehyde ^add^
30	1-Octen-3-one ^add^
31	2-Octanone ^add^
32	Octanal ^add^
33	*trans*,*trans*-2,4-Heptadienal ^add^
34	D(+)-Limonene ^add^
35	Isoamyl butanoate ^MS^
36	1-Octanol ^add^
37	Allyl hexanoate ^MS^
38	2-Nonanone ^istd^
39	Linalool ^add^
40	Isoamyl isovalerate ^MS^
41	L-Menthone ^add^
42	D-Menthone ^add^
43	D/L-Menthol ^add^
44	2-Decanone ^istd^
45	Decanal ^ref^
46	Neral ^add^
47	L-Carvone ^ref^
48	Geranial ^add^
49	trans-Anethole ^ref^
50	Eugenol ^ref^
A–C	other terpenes ^MS^

**Table 3 molecules-27-08067-t003:** The quantification results for the substances found and calibrated for the banana-flavored e-liquid, from a quadruplicate measurement and converted to substance amount in the pure e-liquid.

Substance	Content by Boltzmann Fit/µg/g	Content by Linearization/µg/g
Ethyl acetate	4.9±0.5	3.5±0.2
Isoamyl alcohol	above cal. range	31.8±2.4 *
(+)-α-Pinene	22.9±1.2	20.8±0.8
Octanal	27.3±2.6	17.9±0.5
1-Octanol	2.1±0.3	2.0±0.4
Linalool	14.1±1.5	12.6±1.2
Menthol	40.9±12.9 *	29.5±4.2
Geranial	5.1±0.4	4.7±0.4

^*^ Calculated analyte content partially outside the calibrated range.

## Data Availability

Not applicable.

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
