# Peer review of "Quantitation of Flavor Compounds in Refill Solutions for Electronic Cigarettes Using HS-GCxIMS and Internal Standards"

_molecules, 2022, doi:10.3390/molecules27228067_

Round 1
Reviewer 1 Report
- The objective on the third page and line 92 to 98 is not clear and needs to be rewritten to make it easier for the reader to understand the objective of the research well.
- In the part about the results and their discussion, it was found that the researchers began to discuss the second table before the first, on page p, line 104
- The table that was placed on the fourth page has no title or number. Why?
- Why did the researcher not write the method that was used in the statistical analysis of the results, so I think that it should be written in the materials and methods used in the research.
Author Response
Dear Reviewer,
Thank you for your input and for supporting our manuscript “Quantitation of Flavor Compounds in Refill Solutions for Electronic Cigarettes using HS-GCxIMS and Internal Standards”.
The objective in lines 92 - 101 has been rewritten to clarify the statement on the calculation of LOD and LOQ and why we find this procedure to be useful and better than other approaches.
The table on page four was part of the caption of Figure 1. This misunderstanding and the other reviewer’s comment clearly show this to be not a good idea. Therefore, this table has been added as a separate table (now Table 2) with its own caption and numbering.
The table, previously named Table 2, was positioned as the last table, as it needed to be displayed in a landscape setting and summed up all previously mentioned results, rounding off the results section. Following your suggestion, it has been moved to be the opener of the results section. We know that the positioning will change when the article’s publishing layout is applied.
Unfortunately, we are unsure which statistical analysis we need to elaborate on. We assume you want us to explain the processing of multiple measurements. Results originating from multiple measurements are summed up as the arithmetic mean using the standard deviation as the range or error bar. This is now indicated on lines 382 - 383.
Thanks again for your help and support.
With best regards
Alexander Augustini, on behalf of all authors
Reviewer 2 Report
In my opinion, I recommend that this manuscript be published after minor changes:
I suggest that the authors to consider to display the tabular representation of the compounds in Figure 1 as a separate table and the title below the image change in accordance to that.
Page 5 line 163 typographical error
Page 9 line 275 typographical error the sentence begins with a dot.
Page 15 line 435, a three question marks, I assume that it is typographical errors.
Author Response
Dear Reviewer,
thank you for your input and for supporting our manuscript “Quantitation of Flavor Compounds in Refill Solutions for Electronic Cigarettes using HS-GCxIMS and Internal Standards”.
We have revised the manuscript to correct all errors you have found and added the table as separate item.
Thanks again for your help and support.
With best regards
Alexander Augustini in behalf of all authors